# Hepato-Renal Crosstalk in Acute and Chronic Disease: From Shared Pathways to Therapeutic Targets

**DOI:** 10.3390/biomedicines13071618

**Published:** 2025-07-01

**Authors:** Anna Clementi, Grazia Maria Virzì, Massimiliano Sorbello, Nenzi Marzano, Paola Monciino, Jose Said Cabrera-Aguilar, Giovanni Giorgio Battaglia, Claudio Ronco, Monica Zanella

**Affiliations:** 1Department of Nephrology and Dialysis, Santa Marta and Santa Venera Hospital, 95024 Acireale, Italypaola.monciino@yahoo.it (P.M.); giovanni.giorgio.battaglia@hotmail.it (G.G.B.); 2Department of Nephrology, Dialysis and Transplant, St Bortolo Hospital, 36100 Vicenza, Italy; monica.zanella@aulss8.veneto.it; 3International Renal Resarch Institute Vicenza (IRRIV) Foundation, 36100 Vicenza, Italyjosesaidcabrera@gmail.com (J.S.C.-A.); cronco@goldnet.it (C.R.); 4Department of Anestesia e Rianimazione, Giovanni Paolo II Hospital, 97100 Ragusa, Italy; 5Anesthesia and Intensive Care, Kore University, 94100 Enna, Italy; 6Nephrology Service, Hospital Civil de Guadalajara Fray Antonio Alcalde, Guadalajara 44200, Mexico

**Keywords:** liver-kidney crosstalk, hepatorenal syndrome, liver fibrosis, chronic kidney disease, inflammation, apoptosis, oxidative stress

## Abstract

Hepato-renal crosstalk is a complex biological communication between liver and kidneys mediated by various factors, including cellular, endocrine, and paracrine molecules. This interaction highlights the functional consequences that damage in one organ can have on the other. In particular, the liver and kidney play a pivotal role in maintaining body homeostasis, as they are both involved in the excretion of toxic bioproducts and drugs. The overlap of liver and kidney disease has both therapeutic and prognostic implications. Therefore, a better understanding of the mechanisms involved in the pathogenesis of this bidirectional crosstalk is essential for improving the management of these clinical conditions and patient outcomes. Specifically, a multidisciplinary approach involving hepatologists and nephrologists is crucial to reduce the long-term burden of these clinical settings. This review focuses on the hepato-renal crosstalk in the context of liver and kidney disease, with particular attention to acute kidney injury associated with liver injury, hepatorenal syndrome and, chronic kidney disease in the context of liver fibrosis.

## 1. Introduction

In the human body, the different organ systems are tightly interconnected. Although the concept of “crosstalk” has not yet been precisely defined, the term originates from electronics, referring to any signal/circuit which affects other systems. In molecular biology, the term “crosstalk” indicates conditions where specific components of a signal transduction pathway influence other pathways. Organ crosstalk refers to complex biological communication between distant organs, mediated by different factors, particularly cellular, endocrine, and paracrine molecules. It is necessary for the maintenance of body homeostasis. Sometimes, altered crosstalk may occur due to the structural and/or functional impairment of a malfunctioning organ. These effects are generally negative and injurious for the target organs and, consequently, for the entire body [1,2,3,4]. Therefore, altered crosstalk may trigger multi-organ imbalance by creating a vicious cycle [3,4] (Figure 1).

The liver and kidneys play a pivotal role in maintaining body homeostasis as they are essential for the excretion of toxic bioproducts and drugs [5]. The overlap of liver and kidney disease is so frequent in clinical practice that it supports the hypothesis of a bidirectional organ crosstalk. Various factors involved in liver-kidney crosstalk may contribute to the development of an altered interaction between these two organs, affecting both disease severity and duration. Indeed, the damaged organ can influence the function of distant systems through cell-mediated and humoral pathways, leading to higher therapeutic costs as well as increased morbidity and mortality. The severity of liver injury often parallels the severity of kidney damage, and vice versa [6,7] (Figure 2).

Multiple factors are involved in the pathogenesis of liver and kidney diseases and in their coexistence. Notably, in acute settings, inflammation, oxidative stress, and apoptosis play a pivotal role in the development of acute kidney injury (AKI) associated with liver damage. Moreover, in specific autoimmune diseases, such as systemic lupus erythematosus (SLE), both the liver and kidneys can be target organs of a dysregulated immune response. Additionally, small non-coding RNAs appear to be implicated in the pathogenesis of both liver and kidney disease, representing a possible therapeutic target in these clinical conditions. Hepatorenal syndrome (HRS) and chronic kidney disease (CKD) complicating liver fibrosis both represent clinical conditions characterized by the involvement of the two organs.

### Aim of the Work

A better understanding of hepato-renal crosstalk is essential for improving both the management and outcomes for patients with liver and kidney injury. In the wide spectrum of concomitant liver and renal dysfunction, this review focuses on current knowledge regarding pathological hepato-renal crosstalk, particularly in relation to AKI associated with liver injury, HRS, and CKD in liver fibrosis.

## 2. Material and Methods

A comprehensive literature search was conducted using electronic databases including PubMed, Scopus, and Web of Science. The search strategy combined MeSH terms and free-text keywords related to “organ crosstalk”, “hepato-renal crosstalk,” “liver-kidney interaction,” “acute kidney injury,” “hepatorenal syndrome”, “chronic kidney disease”, and “liver fibrosis”. Only articles published in English were considered. Additional references were identified by screening the bibliographies of selected articles. The inclusion criteria focused on studies exploring the pathophysiological mechanisms, molecular pathways, and clinical implications of liver-kidney interactions in both acute and chronic settings. Reviews, original research articles, and consensus guidelines were included to provide a broad and updated overview of the topic. Case reports and case series were excluded to maintain a focus on studies with broader generalizability and methodological rigor.

## 3. AKI Associated with Liver Injury

In hepato-renal crosstalk, both soluble inflammatory mediators and immune cells play a pivotal role. Pro-inflammatory cytokines and reactive oxygen species (ROS), as well as apoptosis, are involved in the pathogenesis of combined acute kidney-liver injury [8].

### 3.1. Inflammation

Pro-inflammatory cytokines, such as tumor necrosis factor-α (TNF-α), interleukin (IL)-1, IL-6, and IL-10 induce tissue injury and organ impairment. Notably, inflammation seems to play a pivotal role in the triggering and worsening of AKI, and its effects involve not only the kidney. Animal models of AKI have demonstrated that acute renal dysfunction stimulates inflammatory responses in the liver [1]. Specifically, AKI is associated with increased hepatic vascular permeability, through the infiltration of neutrophil and T-lymphocytes [9]. Moreover, current evidence has shown that AKI is related to increased levels of pro- and anti-inflammatory cytokines originating either from enhanced renal production or from reduced clearance [10]. Indeed, AKI is characterized by high levels of IL-6 in tubular epithelial cells, which are then released into circulation and able to activate Kupffer cells. These cells induce the hepatic production of IL-10 [11,12,13] and TNF-α, which also increases in the setting of renal ischemia–reperfusion injury and is responsible for a 10-fold increase in myeloperoxidase (MPO) activity in the liver [14]. Damaged hepatic cells release humoral factors, leading to the activation of the Toll/interleukin-1 receptor superfamily, which, in turn, activate nuclear factor kB (NF-kB) via a conserved domain. Once activated, NF-kB moves into the nucleus, affecting gene expression. Toll-like receptors (TLR) pathways are up-regulated by NF-kB, resulting in increased inflammatory cytokine expression both at intra- and extracellular levels [15,16]. The acute inflammatory response consists of an increase in the release of pro-inflammatory cytokines (TNF-α, IL-1 and IL-6), expression of adhesion molecules, hepatic vascular permeability, and neutrophil and T-lymphocyte infiltration [14]. TNF-α and IL-6 levels are higher both in the serum and in the liver, along with markers of apoptosis and oxidative stress, as well as bilirubin and serum alanine transaminase (ALT) [9,17].

### 3.2. Oxidative Stress

Oxidative stress plays a critical role in the pathogenesis of cellular dysfunction, tissue injury, and organ failure. It characterizes different clinical scenarios in which ischemia and reperfusion injury (IRI) are typical, such as organ transplantation, myocardial infarction, cerebral vascular accidents, hemorrhagic shock, metabolic syndrome, hypertension, diabetes, nephrolithiasis, general cognitive loss, and Alzheimer’s disease [18,19,20,21,22].

A typical feature of oxidative stress is the imbalance between ROS and antioxidant systems of the body. ROS (hydrogen peroxide-H_2_O_2_, superoxide-O_2_^−^, and hydroxyl-OH^−^) are released during cellular metabolic processes [23]. They may stimulate lipid peroxidation (LPX) of cell membranes, leading to cell damage. As a result, antioxidant systems, including glutathione peroxidase (GPx), catalase (CAT), and superoxide dismutase (SOD), are necessary to counteract the effects of ROS [24]. Maintaining cellular homeostasis therefore requires a specific equilibrium between pro-oxidant/antioxidant molecules and ROS. The loss of this redox homeostasis induces an activation of the immune system and the creation of a pro-inflammatory and pro-fibrotic milieu through specific mechanisms that stimulate structural and functional abnormalities [25,26]. The relationship between oxidative stress and inflammatory pathways is not fully understood and may involve heat shock proteins, which could serve as potential targets for therapeutic interventions [27,28].

ROS and reactive nitrogen species (RNS) are normally present in hepatocytes since they are essential for physiological processes, such as oxidative respiration, growth, regeneration, microsomal defense, and apoptosis [29]. When the balance between ROS generation and cellular antioxidant defenses is disrupted, net oxidative stress ensues [29]. In the liver, neutrophils and Kupffer cells release free radicals when stimulated by reactive oxygen species (ROS) and reactive nitrogen species (RNS), through the activation of mitochondria and cytochrome P450 enzymes [30]. The cellular damage induced by oxidative stress involves hepatocytes, endothelial, Kupffer, and stellate cells through pathways of inflammation, ischemia, apoptosis, necrosis, and regeneration [30]. ROS are responsible for the alteration of lipids, protein, and DNA and they are implicated in the pathogenesis of liver disease and IRI [31].

Specifically, in the setting of ischemic AKI, oxidative stress plays a fundamental role in determining liver damage. Indeed, typical features of this clinical condition include an increase in hepatic malondialdehyde (MDA) levels, which represents an index of lipid peroxidation, and higher levels of ALT [32]. At the same time, a decrease in total glutathione (an antioxidant agent) concentrations is generally observed [32].

Serteser et al. [14] observed a decrease in hepatic SOD and CAT activities in cases of kidney IRI, together with reduced liver glutathione concentrations. Moreover, the administration of glutathione before renal ischemia–reperfusion decreased liver injury, as well as MDA and transaminases concentrations [32]. Other authors found an increased hepatic expression of SSAT (spermine/spermidine-N1-acetyl transferase), responsible for the rate-limiting polyamine metabolism, 6 h after kidney IRI or in case of bilateral nephrectomy, thus inducing an increase in reactive oxygen intermediates. Also, in this study, the administration of reduced glutathione (GSH) was demonstrated to markedly ameliorate liver injury [32]. Moreover, Gurley et al. have shown a significant association between increased levels of cytokines and reduced oxidative metabolism mediated by hepatic cytochrome P-450 (CYP) activity [33].

### 3.3. Apoptosis

Apoptosis is a physiological cellular mechanism aimed at maintaining cell populations in tissues in the setting of a programmed pattern both in prenatal development and aging [34]. Apoptosis is also crucial for the clearance of harmful cells, such as tumor and autoreactive immune cells [35].

Different apoptotic pathways are activated in liver disease. The extrinsic pathway is activated by the ligands of the TNF receptor superfamily: TNF-α, Fas ligand (FasL), TNF-related apoptosis inducing ligand (TRAIL) [36,37]. Apoptosis in the liver is generally mediated by the extrinsic pathway due to the ubiquitous expression of death receptors and their ligands in the hepatic cells. This occurs in cholestatic liver disease [38,39], viral hepatitis [40] (B and C) [41,42], nonalcoholic steatohepatitis (NASH) [43] and liver transplantation [44]. The intrinsic pathway is activated in response to cellular stress, DNA damage, radiation, growth serum withdrawal, or other stress signals identified by Bcl-2-homology 3 only (BH3-only) proteins, responsible for an alteration of the outer mitochondrial membrane permeability and potential [45]. The intrinsic pathways are also activated by acetaminophen which is transformed into the toxic metabolite N-acetyl-p-benzoquinone imine by the cytochrome P450 system [46,47]. However, both apoptotic pathways may be triggered in acute liver failure [48] and alcoholic hepatitis [49].

When liver damage becomes chronic, it induces significant changes that affect systemic circulation and kidney perfusion [50]. In hospitalized patients with chronic liver disease, the incidence of AKI is approximately 20% [51], with prerenal AKI being the most common cause, accounting for about 68% of all cases [52]. HRS, which is not volume-responsive, represents approximately 25% of the cases of prerenal AKI, which constitutes only 17% of cases of AKI in hospitalized patients suffering from cirrhosis [52]. Apoptosis also occurs in the kidney, especially in post-transplant acute renal failure, through a genomically mediated process including heat shock proteins [53]. Apoptotic pathways are also activated in toxic and obstructive acute renal failure, cortical atrophy post-papillary necrosis and in chronic renal disease secondary to subtotal nephrectomy [54]. Normal human glomerular cell apoptosis is generally low, and it increases in case of human glomerulonephritis, such as IgA nephropathy, acute postinfectious glomerulonephritis, or lupus nephritis [55,56]. All the cells located in the glomerulus, such as mesangial and endothelial cells, glomerular epithelial cells and podocytes, may undergo apoptosis in kidney disease [57].

Liver injury generally develops in the early stages of kidney injury, and it is characterized by high levels of pro-inflammatory cytokine TNF-α, activated oxidative stress, low antioxidant levels, and, especially in the case of renal failure secondary to bilateral nephrectomy, pronounced apoptosis [32]. Indeed, the examination of hepatocytes has highlighted an increased number of cells with activated caspase-3 staining twenty-four hours after bilateral nephrectomy. The number of apoptotic cells correlates with TNF-α levels, and with the severity and extent of renal injury and liver cell damage. Nevertheless, the signal initiating the apoptotic response is still not completely understood in bilateral nephrectomy. In animal models of IRI or bilateral nephrectomy, significant liver damage is observed, thus indicating that kidney injury per se, regardless of its etiology, can cause hepatic injury. The increased mortality rate observed in hospitalized patients with AKI may be associated, at least in part, to remote multi-organ failure secondary to kidney dysfunction. Indeed, the severity of liver injury is associated with the stage of kidney damage [32]. The role of apoptosis has been suggested by the increased hepatic caspase-3 staining after experimental AKI, as well as increased TUNEL positivity in the peri-portal region [58].

### 3.4. Autoimmune Disease

In the setting of SLE, anti-ribosomal proteins (RibP) antibodies have been associated with organ involvement, particularly with lupus nephritis (LN) and autoimmune hepatitis [59]. In a relatively recent meta-analysis, Choi et al. suggested that the relationship between anti-RibP antibodies and renal disease may be confounded by the tight association between anti-RibP and anti-double stranded DNA (dsDNA) antibodies [60]. Furthermore, titers of anti-RibP antibodies also changed with flares and remission in LN, even though this was also correlated with the levels of anti-dsDNA antibodies. Thus, a combination of anti-dsDNA and anti-RibP appears to be most strongly associated with renal involvement in SLE [60].

A strong association between anti-RibP and lupoid hepatitis (LH) has also been observed. Anti-RibP may play a pivotal role in the pathogenesis of lupoid hepatitis since they are able to penetrate live human hepatoma tissue culture cells, inducing morphological changes [61]. Liver dysfunction affects up to 50% of SLE patients due to drugs, viruses and autoimmune hepatitis [62]. However, it is estimated that patients with SLE with no other obvious causes of liver dysfunction develop hepatitis in 28–42% [63]. Furthermore, the presence of anti-RibP has also been related to more severe forms of hepatitis [64].

## 4. Small Non-Coding RNAs: A Possible New Entity Implicated in Organ Crosstalk

Small non-coding RNAs (~18 to 30 nucleotides) represent a specific mechanism of gene regulation. They are post-transcriptional regulators that bind to 3′-untranslated regions (3′UTR) of target messenger RNAs (mRNAs) [65]. Recent literature highlights the regulatory roles of these RNAs, like transcriptional gene silencing/activation and post-transcriptional gene silencing [2,66].

A growing number of studies on small RNAs have demonstrated a new association of these non-coding molecules with different biological functions [67]. Small RNAs seem to play a central role in basic regulation processes, such as cellular differentiation, growth/proliferation, apoptosis/cell death, migration, metabolism, and defense [65,68,69,70]. Therefore, small RNAs could be considered key regulators in development, physiology, and disease states [68,71]. Small RNAs are classified into three main categories: microRNAs (miRNAs), small interfering RNA (siRNAs), and Piwi-interacting RNAs (piRNAs), based on their origin, structure, associated effector proteins, and biological functions [72,73]. In particular, miRNA dysregulation has also been found to be related to a wide spectrum of diseases, most notably cancer [70,71,74,75].

Recent studies have provided evidence that miRNAs are abundant in the liver, where they modulate a range of hepatic functions [76]. As such, the deregulation of miRNA expression may represent a key factor in various liver diseases, such as viral hepatitis, hepatocellular carcinoma, polycystic liver disease, and liver IRI.

Lee et al. reported that the cholangiocyte cell line (CCL) derived from polycystic kidney disease rats (PCK) displayed marked changes in miRNA expression compared with normal rat cholangiocytes [77]. In particular, the authors found decreased levels of miR-15a in both PCK-CLL cells and in liver tissues from PCK and patients with polycystic liver disease. This finding was associated with upregulation of the cell-cycle regulator cell division cycle 25 A (Cdc25A), responsible for cell proliferation and cyst growth [77]. These data suggest that suppression of miR-15a activity is involved in hepatic cystogenesis through the dysregulation of Cdc25A [77].

Cilia and flagella are ancient and conserved organelles playing different roles, including whole-cell locomotion, chemosensation, photosensation, and sexual reproduction [78]. Several diseases are related to ciliary dysfunction, including polycystic kidney disease (PKD) [79]. Pandey et al. explored the possible role of miRNAs in PKD and found that 30 miRNAs were differentially regulated in PKD rats [80]. Notably, miR-15a and miR-17 are involved in the pathogenesis of cystic kidney disease [81,82,83]. In addition, miR-192 and miR-377 [84] are implicated in the matrix deposition [81], while miR-200 [85,86,87] and miR-205 [88] are related to epithelial-to-mesenchymal transition [81]. MiR-15a may be involved both in polycystic liver and kidney disease.

In the setting of liver IRI, miRNA-182-5p has been shown to attenuate hepatic damage by directly suppressing TLR4 and subsequently inhibiting the production of several inflammatory cytokines, such as TNF-α and IL-6 [89]. Moreover, miRNA-182-5p seems to be consistently up-regulated after ischemic insults in kidneys of both human and animal models. Selective in vivo inhibition of miRNA-182-5p in IRI models has also improved kidney function [90]. Furthermore, miRNA-182-5p is capable of reducing the expression of the anti-apoptotic protein BCL2 and cell-cycle regulatory cyclin-dependent kinase inhibitors (p21), thereby inhibiting cell regeneration [91]. Therefore, miRNA-182-5p is involved in both liver and kidney IRI.

Table 1 summarizes the principal molecular factors involved in hepato-renal crosstalk.

## 5. Hepatorenal Syndrome

Hepatorenal syndrome (HRS) often complicates the advanced stages of cirrhosis, being characterized by renal failure and severe alterations in body circulation. Specifically, kidney dysfunction appears to be related to the intense renal vasoconstriction, secondary to splanchnic vasodilatation [92,93].

The diagnosis of HRS requires the exclusion of other causes of renal failure. Renal blood flow is decreased due to an alteration of the systemic circulation. Even though HRS generally occurs in the setting of cirrhosis, it may also develop in other liver diseases, such as alcoholic hepatitis and acute liver failure [94,95].

HRS is classified in two different types. Type 1 HRS is characterized by the rapid development of renal failure, which represents its main clinical feature. By contrast, in Type 2 HRS, the degree of renal dysfunction is less severe and more stable over time [96]. According to a definition proposed by Salerno et al. [97], Type 2 HRS may be considered a functional renal disorder with a serum creatinine level above 1.5 mg/dL, developing in the setting of cirrhosis and ascites, in the absence of shock, nephrotoxic drugs, or intrinsic kidney disease. In contrast, in patients with Type 1 HRS, functional renal disorder progresses rapidly, with a serum creatinine doubling from the baseline and exceeding 2.5 mg/dL within two weeks [97]. In patients with cirrhosis and Type 1 or Type 2 HRS, median survival has been reported as approximately 1 and 6.7 months, respectively [98].

Recently, a new classification system for hepatorenal disorders in cirrhosis has been proposed by the Acute Dialysis Quality Initiative (ADQI) and International Ascites Club (IAC) [99,100]. According to this classification, Type 1 HRS is defined as a specific form of AKI, while Type 2 HRS is not classified as chronic kidney disease [99]. There is an ongoing debate regarding the classification of Type 2 HRS. Some experts do not consider it a structural chronic kidney disease (CKD), but rather a functional renal disorder [99]. Moreover, in some patients with cirrhosis, reduced renal blood flow with a normal or low–normal glomerular filtration rate (GFR) is observed, without meeting the criteria of HRS in its current definition. This group of cirrhotic patients has not yet been included in any HRS classification [101].

### Pathophysiology of HRS

HRS generally develops in the advanced stages of hemodynamic dysfunction, which begins early during liver disease, even before ascites become medically detectable. The characteristic hemodynamic alteration changes worsen as cirrhotic patients progress from the pre-ascitic stage to diuretic-sensitive ascites, diuretic-resistant ascites, and, ultimately, HRS. Multiple factors contribute to the pathogenesis of HRS, including splanchnic vasodilation, decreased effective arterial blood volume, a hyperdynamic circulatory state characterized by elevated cardiac output (CO) and reduced systemic vascular resistance, vasoconstriction of extrasplanchnic vascular beds such as the renal and cerebral circulation, and activation of compensatory systems, like the renin–angiotensin–aldosterone system (RAAS), the sympathetic nervous system (SNS), and non-osmotic vasopressin release [102]. Arterial vasodilatation primarily involves splanchnic circulation through the release of potent vasodilators, such as nitric oxide (NO) [103]. In cirrhotic patients, nitric oxide (NO) production is elevated in the splanchnic circulation as a result of shear stress-induced upregulation of endothelial NO synthase (eNOS) activity and activation of eNOS mediated by endotoxins [104,105]. Additional vasodilatory mediators—including calcitonin gene-related peptide (CGRP), substance P, carbon monoxide, endocannabinoids, and adrenomedullin—may also contribute to splanchnic vasodilation. The increased activity of vasoconstrictor systems is responsible for the marked reduction in renal perfusion and glomerular filtration rate (GFR), while tubular function is preserved.

All these changes induce intense renal vasoconstriction with a subsequent reduction in GFR. Moreover, increased retention of sodium (via renin-angiotensin and sympathetic nervous system) and free water (via arginine vasopressin) are observed in advanced cirrhosis [103,104,105,106]. In the early phases of decompensated cirrhosis, renal perfusion is maintained within the normal range due to the increased release of vasodilatory factors (e.g., prostaglandins) [107]. In the advanced phases of the disease, renal perfusion can no longer be maintained due to the severe arterial underfilling, which causes maximal activation of vasoconstrictor systems and decreased production of renal vasodilators. The splanchnic area escapes the effect of vasoconstrictors, probably due to the greatly enhanced local production of vasodilators.

In the kidneys, vasoconstriction is normally offset by enhanced production of vasodilatory prostaglandins. However, in HRS, both urinary prostaglandin levels and renal medullary cyclooxygenase activity are diminished—a pattern not commonly seen in other forms of AK)—suggesting that impaired renal prostaglandin release contributes to HRS pathogenesis [108].

In both human and animal studies, increased renal SNS activity has been shown to contribute to the renal vasoconstriction observed in HRS. In humans, transjugular intrahepatic portosystemic shunt (TIPS) insertion improves renal blood flow and reduces SNS activity, which subsequently returns after TIPS occlusion [109,110].

A hyperdynamic circulatory state, marked by elevated CO and heart rate, is a hallmark of cirrhosis. Cirrhotic cardiomyopathy in these patients is characterized by impaired systolic and diastolic responses to stress, often accompanied by ventricular hypertrophy or chamber dilation [111,112]. As a result, renal perfusion decreases and renal vasoconstriction develops, particularly after infections or other stressors that precipitate HRS. Several factors influence cardiac function in this context: (1) myocardial hypertrophy and fibrosis driven by neurohumoral activation; (2) impaired β-adrenergic receptor signaling due to prolonged sympathetic nervous system (SNS) stimulation; and (3) the suppressive effects of circulating cytokines—such as tumor necrosis factor-α (TNF-α) and nitric oxide (NO)—on ventricular performance [113,114].

It has also been observed that during episodes of hepatic failure decompensation, a state of relative adrenal insufficiency may occur, reflected by an inadequate cortisol response to adrenocorticotropic hormone (ACTH). This, combined with the impairment of compensatory systems, leads to a greater risk of severe complications such as sepsis, HRS, and death [115]. Currently, the treatment of HRS with acute kidney injury is based on vasopressor use and shows benefit when an improvement in mean arterial pressure (MAP) is achieved [116]. It has been observed that bilirubin levels ≥10 mg/dL are an independent predictor for non-response to terlipressin—the vasopressor of choice in this context [117]. This suggests that recovery of both organs follows a more bidirectional process than previously thought.

Bacterial infections, particularly spontaneous bacterial peritonitis (SBP), represent the primary trigger for hepatorenal syndrome (HRS). Despite appropriate treatment and resolution of the infection, HRS occurs in 20–30% of patients with SBP [118,119]. Patients with spontaneous bacterial peritonitis (SBP) who have pre-existing hyponatremia, increased serum creatinine, or elevated plasma or ascitic cytokine levels at diagnosis are more likely to develop HRS. This increased risk is primarily due to the inflammatory response triggered by SBP, which plays a central role in the development of HRS [119,120].

Table 2 reports the factors involved in the pathogenesis of HRS.

## 6. Chronic Kidney Disease and Liver Fibrosis

Even though the relationship between the stage of liver fibrosis and the incidence of chronic kidney disease (CKD) is not yet fully understood, these two clinical conditions share common risk factors and pathological pathways, in particular hypertension, dyslipidemia, insulin resistance, and obesity [121].

Non-alcoholic fatty liver disease (NAFLD) has been demonstrated to be related to a higher risk of developing CKD [122]. Seo et al. have found that this relationship may go beyond metabolic syndrome as the sole cause of both diseases separately. According to their study, advanced liver fibrosis in patients with NAFLD is independently associated with an increased risk of developing chronic kidney disease in patients with type 2 diabetes, compared to patients with type 2 diabetes and NAFLD who have not yet developed fibrosis [123].

A recent meta-analysis of 13 longitudinal studies has shown a moderately higher risk of incident CKD stage ≥ 3 in NAFLD over a follow-up period of 9.7 years, independently of age, sex, hypertension, obesity, and type 2 diabetes mellitus [124]. Interestingly, the probability of developing CKD is higher in more severe stages of liver disease, specifically with worsening hepatic fibrosis [124]. Lee et al. have demonstrated that agile 3+ and agile 4 are reliable indicators for identifying NAFLD patients at high risk of developing CKD. Moreover, early glycemic control in the prediabetic stage might have a potentially renoprotective role in these patients [125].

NAFLD and CKD represent two global diseases affecting almost 30% and 10–15% of the general population worldwide, respectively [126]. Both diseases are associated with a reduced quality of life, premature mortality, and high healthcare costs [126,127]. Since NAFLD is strongly linked to insulin resistance, obesity and type 2 diabetes mellitus, in 2020 a group of international experts proposed the terms metabolic dysfunction-associated fatty liver disease (MAFLD) and metabolic dysfunction-associated steatotic liver disease (MASLD) to replace the old one NAFLD [128,129].

Currently, there is no literature on the association between MASLD and the probability of developing CKD [130]. Instead, MAFLD has been demonstrated to increase the risk of incident CKD in a 10-year follow-up, even after adjusting for age, sex, dyslipidemia, hypertension, and obesity [131]. Moreover, the coexistence of MAFLD and CKD may also increase the probability of cardiovascular disease more accurately than CKD and MAFLD alone [124].

In the pathogenesis of CKD in the context of NAFLD/MASLD, a complex interplay of hemodynamic and metabolic factors appears to play a central role.

RAAS plays a crucial role in the pathogenesis of both NAFLD and CKD. In the liver, angiotensin II increases insulin resistance, lipogenesis, and the release of pro-inflammatory cytokines, such as IL-6 and transforming growth factor–beta (TGF-β) [132,133]. IL-6 is responsible for renal fibrosis and endothelial dysfunction, in addition to tubulointerstitial inflammation and chronic injury through leukocyte recruitment and NF-kB activation [134]. In the kidneys, RAAS activation leads to abnormal lipid accumulation, increased oxidative stress, and inflammation, thus contributing to glomerulosclerosis and CKD progression [135].

Moreover, it is suggested—although not yet conclusively validated—that subclinical portal hypertension in non-cirrhotic NAFLD could contribute to a subclinical hepatorenal reflex, which, over time, promotes the development of HRS and progression of chronic kidney disease [136].

Atherogenic dyslipidemia, which is an important factor in the pathogenesis of kidney damage, represents another key feature of NAFLD, where increased small, low-density lipoprotein (LDL) cholesterol, low high-density lipoprotein (HDL) cholesterol, and high triglycerides are typical [137,138,139].

Furthermore, high levels of pro-coagulant and pro-fibrogenic factors in NAFLD may worsen vascular damage and accelerate atherosclerosis in CKD [137,138,139]. In NAFLD with severe fibrosis, the activation of NF-kB pathway can induce the release of pro-inflammatory molecules, thus leading to CKD [137,140].

Other factors in MASLD, such as chronic hyperglycemia, hypertension, and abdominal obesity, may lead to the progression of macro- and microvascular renal complications, thereby contributing to the pathogenesis of CKD [137]. All these metabolic conditions induce renal oxidative stress and activate pro-inflammatory immune cells, which may promote albuminuria and a progressive reduction in estimated glomerular filtration rate (eGFR) [141].

It has been observed that certain bile acids activate two receptors in the kidney that may play a crucial role in the liver-kidney crosstalk: the nuclear receptor farnesoid X receptor (FXR) and the membrane-bound G protein-coupled bile acid receptor 1 (GPBAR1, also known as TGR5). The activation of FXR and TGR5 reduces renal inflammation, oxidative stress, and fibrosis, suggesting a protective role in kidney disease. In diabetes and obesity (which are bidirectionally linked to NAFLD), FXR and TGR5 are downregulated, and their activation has been proposed as a potential treatment to slow the progression of kidney disease. In the context of HRS, both bile acid metabolism and the expression of FXR and TGR5 receptors are altered. The fact that TGR5 mRNA can be found in liver tissue strongly suggests its activity as an intercellular signaling molecule [142].

Other molecules which may play a role in the pathogenesis of CKD in MASLD are the hepatokines, in particular fibroblast growth factor-21 (FGF-21), whose levels increase in patients with type 2 diabetes mellitus [143], CKD [144] and NAFLD/MASLD [145]. Higher levels of FGF-21 seem to be an adaptive response to metabolic alterations, such as hyperglycemia and insulin resistance [146], even though its renal effects remain unclear.

In addition, recent studies have reported a potential role of some specific genetic polymorphisms, especially rs738409 C>G p.I148M in the *PNPLA3* (patatin-like phospholipase domain-containing protein 3) gene, in the progression of NAFLD [147], MAFLD [148], and CKD [149].

Both NAFLD and CKD are associated with a higher risk of cardiovascular events [150]. Specific metabolic factors which are involved in the pathogenesis of both conditions include insulin resistance, ectopic fat deposition, and activation of the transforming growth factor-β pathway [151,152]. Interestingly, adipose tissue dysfunction seems to contribute to the worsening of CKD in MASLD. Indeed, the inability of adipose tissue to suppress lipolysis leads to the ectopic lipid deposition in the liver and kidneys, typically in the perirenal space, renal sinus, and parenchyma. Renal vasculature is compressed, resulting in increased hydrostatic pressure, renin release, and reduced eGFR [153]. Moreover, renal sinus fat accumulation may induce kidney inflammation and fibrosis, contributing to CKD progression.

Also, gut microbiome alterations play a role in the pathogenesis of NAFLD and CKD [154]. Indeed, increased fructose intake and reduced vitamin D levels may be responsible for a low-grade inflammatory state involved in the development of NAFLD and CKD [154]. The role of short-chain fatty acids (SCFAs), produced by the intestinal microbiota, should be highlighted as key modulators of the inflammatory response and renal cell metabolism. SCFAs interact with specific receptors in the kidney, such as Gpr41, Gpr43, Gpr109a, and Olfr78, regulating physiological processes, such as cell proliferation, apoptosis, and histone inhibition. During AKI, the reduction in beneficial bacteria and the increase in pathogens alter SCFA production, promoting a pro-inflammatory environment that exacerbates kidney damage. Needless to say, liver disease leads to a chronic state of dysbiosis, perpetuating the pathogenic process and predisposing to worse outcomes [155].

Based on the evidence discussed, patients with NAFLD should undergo screening for CKD and renal function should be carefully monitored. Patients with NAFLD and CKD often meet the criteria for cardiovascular-kidney-metabolic (CKM) syndrome, a disorder characterized by different risk factors, such as obesity, insulin resistance, inflammation, and endothelial dysfunction [156]. This highlights the need for integrated screening and management of these patients.

Figure 3 reports the factors involved in the pathogenesis of NAFLD/MAFLD/MASLD and CKD.

## 7. Conclusions

Liver-kidney crosstalk is clinically relevant, and this review analyzes the most important mechanisms involved in the alteration of this organ communication, providing new tools for the diagnosis and treatment of these patients. The management of liver-kidney disease and HRS subtypes represents a great challenge because of the complexity of all the pathophysiological interactions between these two organs. Indeed, a multidisciplinary approach is recommended.

Inflammatory processes, oxidative stress, and apoptosis have been reported as important factors implicated in AKI associated with liver injury, as well as the role of small RNA in liver-kidney crosstalk. The pathogenesis of HRS and the association between CKD and liver fibrosis have been investigated to provide a better understanding of both acute and chronic settings.

Future studies on pathogenetic mechanisms involved in hepato-renal crosstalk are necessary to further improve diagnostic tools and therapeutic strategies in this patient population.

## Figures and Tables

**Figure 1 biomedicines-13-01618-f001:**
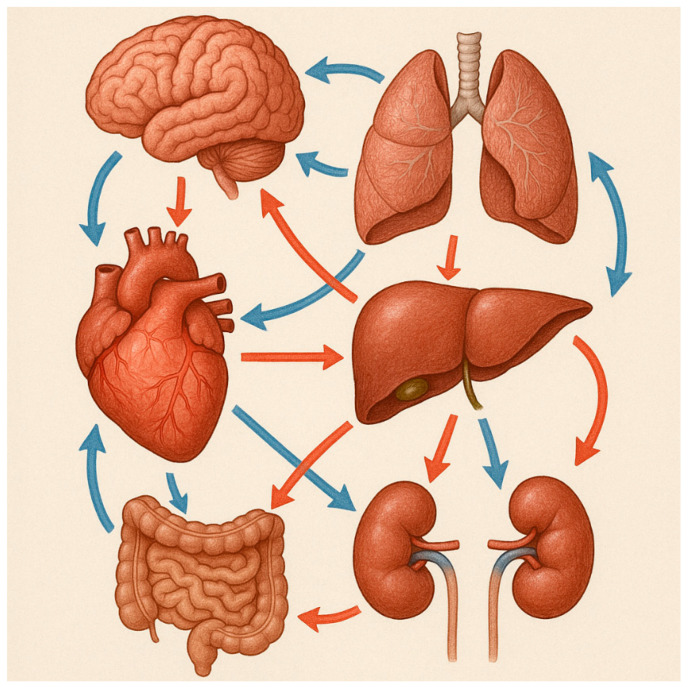
Illustrative example of organ crosstalk among different organs. Organ crosstalk plays an important role in both physiological and pathological conditions.

**Figure 2 biomedicines-13-01618-f002:**
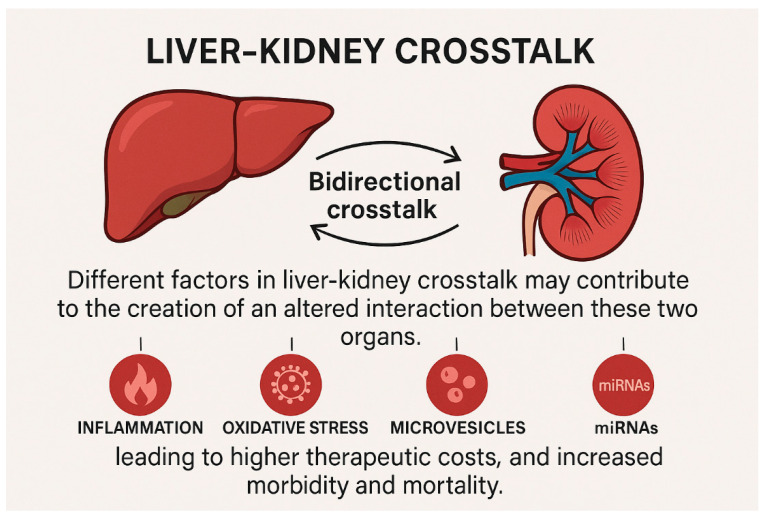
Illustrative example of the bidirectional organ crosstalk among liver and kidneys. The damaged organ can influence the function of distant systems through cell-mediated and humoral pathways, leading to higher therapeutic costs, and increased morbidity and mortality.

**Figure 3 biomedicines-13-01618-f003:**
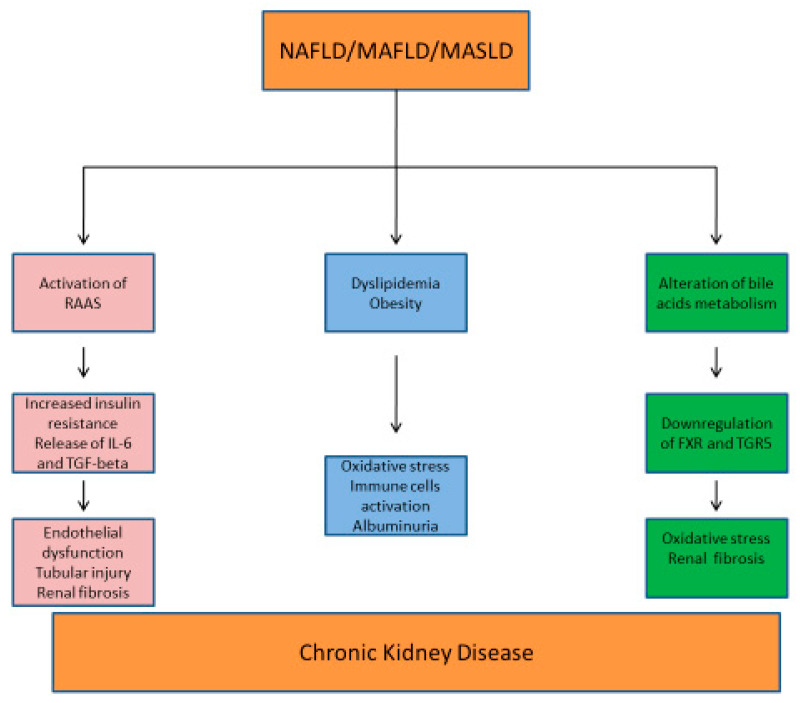
Metabolic mechanisms involved in the pathogenesis of NAFLD/MAFLD/MASLD and CKD. RAAS: renin–angiotensin–aldosterone system; IL-6: interleukin-6; TGF-beta: transforming growth factor; FXR: farnesoid X receptor; TGR5: membrane-bound G protein-coupled bile acid receptor 1.

**Table 1 biomedicines-13-01618-t001:** Key molecular players in hepato-renal crosstalk: interleukin-6 (IL-6); tumor necrosis factor-α (TNF-α); myeloperoxidase (MPO); acute kidney injury (AKI); reactive oxygen species (ROS); reactive nitrogen species (RNS); malondialdehyde (MDA).

	Molecules	Effects
Inflammation	IL-6 release by tubular epithelial cells in the setting of AKI	Activation of Kupffer cells with consequent increase in IL-10, TNF-α and MPO levels in the liver. Increased hepatic vascular permeability, neutrophil and T-lymphocyte infiltration.
Oxidative stress	ROS and RNS in the setting of ischemic AKI	Release of free radicals by neutrophils, Kupffer cells. Alterations of lipids (increased MDA), proteins and DNA.
Apoptosis	TNF-α release in case of bilateral nephrectomy	Caspase-3 activation in hepatocytes with consequent liver injury.
Small non-coding RNA	miR-15a	Involvement in the pathogenesis of liver and kidney cystic disease.
Small non-coding RNA	miRNA-182-5p	Attenuation of liver injury in case of IR. Inhibition of cellular regeneration in case of kidney IR.

**Table 2 biomedicines-13-01618-t002:** Factors involved in the pathogenesis of HRS.

	Effects	Molecular Factors
Hemodynamic alterations	Splanchnic vasodilation and reduced systemic vascular resistance, with reduced effective arterial blood volume	Nitric oxide, calcitonin gene-related peptide, substance P, carbon monoxide, endocannabinoids and adrenomedullin
Renal changes	Renal Vasoconstriction with consequent Reduced glomerular filtration rate	Reduced effective arterial blood volume
	Increased retention of sodium	Increased activity of renin–angiotensin–aldosterone system and of sympathetic nervous system
	Increased retention of water	Increased release of vasopressin

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
