# Peer review of "Hepato-Renal Crosstalk in Acute and Chronic Disease: From Shared Pathways to Therapeutic Targets"

_biomedicines, 2025, doi:10.3390/biomedicines13071618_

Round 1

Reviewer 1 Report

Comments and Suggestions for Authors

Dear authors,

The crosstalk between the liver and kidney axis is highly relevant. Nevertheless, I have some suggestions regarding your review.

Major:

  1. Please rewrite the Abstract section with a clear aim for your review and enhance its scientific soundness.
  2. In my view, the "Introduction" chapter should also briefly describe small non-coding RNAs and their relevance in the context of studying liver-kidney crosstalk. Please provide a detailed outline of your review in the Introduction section, including a brief description of the importance of each chapter in the context of liver-kidney crosstalk.
  3. Please describe the criteria for reviewing the articles, including the databases used, the articles included and excluded in your review, the countries of origin, and the rationale behind the selection of specific articles, etc. So, please include the Material and Methods section.
  4. In the chapter "Inflammation," please clearly clarify the beginning of the relationship: specify the changes in the liver that result from kidney damage, and, conversely, the changes in the kidneys that result from liver damage. Additionally, there is a lack of specifics. For example, where exactly does the increase of IL-6 levels occur in the kidneys (line 72) — in the epithelial cells of the tubules, endothelial cells, or podocytes?
  5. The chapter on oxidative stress in the context of the crosstalk between liver and kidney damage is not entirely clear. In my opinion, it should be rewritten, with the main clarification being how exactly (through what mechanisms) oxidative stress in the liver (and what can lead to it? What are the causes/pathologies/diseases?) affects the condition of the kidneys, and vice versa. Does it mean that if there is oxidative stress in one of these two organs, there will also be oxidative stress in the other?
  6. It is also unclear whether there is a group of small non-coding RNAs common to both liver and kidney damage, or if certain small non-coding RNAs trigger a cascade between related kidney and liver pathologies.
  7. The meaning of Table 1 is also unclear. Please indicate which markers are characteristic of each organ and what their implications are for the other organ.
  8. Please specify in Table 2 in more detail: to which organ the "changes" refer, to which organ the "category" refers, and to which organ the "molecules" refer. It would also be beneficial to indicate which cells produce these "molecules" and which cells they influence, as well as how these "molecules" were measured in clinical and experimental settings (e.g., biochemistry, histology, PCR, Western Blot, etc.).
  9. The chapter "Chronic Kidney Disease and Liver Fibrosis" significantly lacks specific cascades of mechanisms and the sequence of events. Please pay more attention to this and revise Figure 1 to make it more detailed.
  10. Please rewrite the "Conclusion" chapter. For example, there is no need to mention proteomic and genetic studies. Also, please emphasize the relevance and importance of your review. Summarize the key signaling pathways that may be useful for the clinical assessment of patients at risk of developing combined liver and kidney pathology.

Minor:

  1. Please check all typographical errors.
  2. Please add references to the tables within the text.

Author Response

Responses to Reviewer 1

Manuscript ID: biomedicines-3644192
Title: Hepato-Renal Crosstalk in Chronic Disease: From Shared Pathways to Therapeutic Targets

Dear Editors,

We appreciate the time and effort invested in your work. Thank you for your insightful observations. We have addressed comments and recommendations in detail in the attached file (we revised all paper). Thank you for your kind suggestions which have improved the quality of our paper.

The responses are provided below.           

Major:

  1. Please rewrite the Abstract section with a clear aim for your review and enhance its scientific soundness.

The Abstract section has been modified as suggested.

  1. In my view, the "Introduction" chapter should also briefly describe small non-coding RNAs and their relevance in the context of studying liver-kidney crosstalk. Please provide a detailed outline of your review in the Introduction section, including a brief description of the importance of each chapter in the context of liver-kidney crosstalk.

The chapter has been modified as suggested.

  1. Please describe the criteria for reviewing the articles, including the databases used, the articles included and excluded in your review, the countries of origin, and the rationale behind the selection of specific articles, etc. So, please include the Material and Methods section.

The Material and Methods section has been added as suggested.

  1. In the chapter "Inflammation," please clearly clarify the beginning of the relationship: specify the changes in the liver that result from kidney damage, and, conversely, the changes in the kidneys that result from liver damage. Additionally, there is a lack of specifics. For example, where exactly does the increase of IL-6 levels occur in the kidneys (line 72) — in the epithelial cells of the tubules, endothelial cells, or podocytes?

We have modified the chapter as suggested.

  1. The chapter on oxidative stress in the context of the crosstalk between liver and kidney damage is not entirely clear. In my opinion, it should be rewritten, with the main clarification being how exactly (through what mechanisms) oxidative stress in the liver (and what can lead to it? What are the causes/pathologies/diseases?) affects the condition of the kidneys, and vice versa. Does it mean that if there is oxidative stress in one of these two organs, there will also be oxidative stress in the other?

The chapter has been modified as suggested.

  1. It is also unclear whether there is a group of small non-coding RNAs common to both liver and kidney damage, or if certain small non-coding RNAs trigger a cascade between related kidney and liver pathologies.

In this section, small non-coding RNAs involved both in liver and kidney disease have been analyzed and discussed.

  1. The meaning of Table 1 is also unclear. Please indicate which markers are characteristic of each organ and what their implications are for the other organ.

The table has been modified as suggested.                     

  1. Please specify in Table 2 in more detail: to which organ the "changes" refer, to which organ the "category" refers, and to which organ the "molecules" refer. It would also be beneficial to indicate which cells produce these "molecules" and which cells they influence, as well as how these "molecules" were measured in clinical and experimental settings (e.g., biochemistry, histology, PCR, Western Blot, etc.).

            The table has been modified.

  1. The chapter "Chronic Kidney Disease and Liver Fibrosis" significantly lacks specific cascades of mechanisms and the sequence of events. Please pay more attention to this and revise Figure 1 to make it more detailed.

            The chapter and the figure have been modified as suggested.

  1. Please rewrite the "Conclusion" chapter. For example, there is no need to mention proteomic and genetic studies. Also, please emphasize the relevance and importance of your review. Summarize the key signaling pathways that may be useful for the clinical assessment of patients at risk of developing combined liver and kidney pathology.

           The chapter has been rewritten as suggested.

Minor:

  1. Please check all typographical errors.
  2.  

Typographical errors have been checked.

  1. Please add references to the tables within the text.

References to the tables have been added within the text.

Thank you once again to the editors for their kind comments. We hope that the revisions made to the manuscript meet the necessary requirements to proceed with its publication.

Sincerely.

Reviewer 2 Report

Comments and Suggestions for Authors

The review article titled “Hepato-Renal Crosstalk in Chronic Disease: From Shared Pathways to Therapeutic Targets” summarized the clinical relevance between liver diseases and kidney dysfunctions, as well as the molecular pathways known to be involved in such connections. This topic is appealing to readers interested in both kidney and liver diseases. Generally speaking, the review is well written, but the authors could improve the quality of this paper from the following aspects:

  1. More graphic illustrations should be used for the explanation of molecular mechanisms underlying hepato-renal crosstalk. For most mechanisms, certain types of metabolites or immune factors circulating via blood are involved, and the authors should emphasize this point in the figures, considering that the authors did not emphasize the roles of circulation system during the process of describing the molecular mechanism.
  2. For some patients with autoimmune diseases, they also tend to show syndromes of hepatitis and nephritis. The author could cover this topic in this review. For these patients, the dysregulation of immune system includes not only imbalanced cytokines, but also unrestricted immune recognition.
  3. Some up-to-date references are not cited in this review. The authors should try to include more researches published in the recent 5 years. For example: Sun C, Goh GB, Chow WC, et al. Prevalence and risk factors for impaired renal function among Asian patients with nonalcoholic fatty liver disease. Hepatobiliary Pancreat Dis Int. 2024 Jun;23(3):241-248. doi: 10.1016/j.hbpd.2023.08.004. Epub 2023 Aug 3. PMID: 37620227.

The manuscript also has some minor flaws. For example, the abbreviation of AKI should appear with its full name at its first appearance. A few references are still listed by their PMIDs instead of standardized numbers.

Author Response

Responses to Reviewer 2

Manuscript ID: biomedicines-3644192
Title: Hepato-Renal Crosstalk in Chronic Disease: From Shared Pathways to Therapeutic Targets

Dear Editors,

We appreciate the time and effort invested in your work. Thank you for your insightful observations. We have addressed comments and recommendations in detail in the attached file (we revised all paper). Thank you for your kind suggestions which have improved the quality of our paper.

The responses are provided below.

The responses are provided below. 

  1. More graphic illustrations should be used for the explanation of molecular mechanisms underlying hepato-renal crosstalk. For most mechanisms, certain types of metabolites or immune factors circulating via blood are involved, and the authors should emphasize this point in the figures, considering that the authors did not emphasize the roles of circulation system during the process of describing the molecular mechanism.

Two figures have been added.

  1. For some patients with autoimmune diseases, they also tend to show syndromes of hepatitis and nephritis. The author could cover this topic in this review. For these patients, the dysregulation of immune system includes not only imbalanced cytokines, but also unrestricted immune recognition.

A specific chapter on hepatitis and nephritis in SLE has been added.

  1. Some up-to-date references are not cited in this review. The authors should try to include more researches published in the recent 5 years. For example: Sun C, Goh GB, Chow WC, et al. Prevalence and risk factors for impaired renal function among Asian patients with nonalcoholic fatty liver disease. Hepatobiliary Pancreat Dis Int. 2024 Jun;23(3):241-248. doi: 10.1016/j.hbpd.2023.08.004. Epub 2023 Aug 3. PMID: 37620227.

The reference has been added, as the following ones.

  • Choi MY, FitzPatrick RD, Buhler K, Mahler M, Fritzler MJ. A review and meta-analysis of anti-ribosomal P autoantibodies in systemic lupus erythematosus. Autoimmun Rev. 2020 Mar;19(3):102463.

  • Kadatane SP, Satariano M, Massey M, Mongan K, Raina R. The Role of Inflammation in CKD. Cells. 2023 Jun 7;12(12):1581.

  • Ndumele CE, Rangaswami J, Chow SL, Neeland IJ, Tuttle KR, Khan SS, Coresh J, Mathew RO, Baker-Smith CM, Carnethon MR, Despres JP, Ho JE, Joseph JJ, Kernan WN, Khera A, Kosiborod MN, Lekavich CL, Lewis EF, Lo KB, Ozkan B, Palaniappa LP, Patel SS, Pencina MJ, Powell-Wiley TM, Sperling LS, Virani SS, Wright JT, Rajgopal Singh R, Elkind MSV; American Heart Association. Cardiovascular-Kidney-Metabolic Health: A Presidential Advisory From the American Heart Association. Circulation. 2023 Nov 14;148(20):1606-1635.

  1. The manuscript also has some minor flaws. For example, the abbreviation of AKI should appear with its full name at its first appearance.

Acute kidney injury has been added as suggested.

  1. A few references are still listed by their PMIDs instead of standardized numbers.

The references have been corrected.

Thank you once again to the editors for their kind comments. We hope that the revisions made to the manuscript meet the necessary requirements to proceed with its publication.

Sincerely.

Round 2

Reviewer 1 Report

Comments and Suggestions for Authors

Dear authors,

Thank you for addressing all of my comments. At this moment, I would like to draw your attention to just two points.

  1. It might be better to rephrase the sentence on line 32: "In the human body, the different organ systems are tightly interconnected."
  2. The sentence on lines 151-153 and line 301 might need rephrasing, as placing cells, mitochondria, and cytochromes on the same level may not be entirely accurate.

Author Response

REV 1.2 MINOR REV

Manuscript ID: biomedicines-3644192

Title: Hepato-Renal Crosstalk in Chronic Disease: From Shared Pathways to Therapeutic Targets

Dear Editors,

We appreciate the time and effort invested in your work. Thank you for your insightful observations. We have addressed comments and recommendations in detail in the attached file. Thank you for your kind suggestions which have improved the quality of our paper.

The responses are provided below.

  1. It might be better to rephrase the sentence on line 32: "In the human body, the different organ systems are tightly interconnected.

We have modified the sentence (purple).

1. The sentence on lines 151-153 and line 301 might need rephrasing, as placing cells, mitochondria, and cytochromes on the same level may not be entirely accurate.

We have modified these sentences (purple).

Thank you once again to the editors for their kind comments. We hope that the revisions made to the manuscript meet the necessary requirements to proceed with its publication.

Sincerely.